# A Novel Deep Convolutional Neural Network Combining Global Feature Extraction and Detailed Feature Extraction for Bearing Compound Fault Diagnosis

**DOI:** 10.3390/s23198060

**Published:** 2023-09-24

**Authors:** Shuzhen Han, Pingjuan Niu, Shijie Luo, Yitong Li, Dong Zhen, Guojin Feng, Shengke Sun

**Affiliations:** 1School of Mechanical Engineering, Tiangong University, Tianjin 300387, China; hanshuzhen@tiangong.edu.cn; 2Office of the Cyberspace Affairs, Tiangong University, Tianjin 300387, China; 3School of Electronics and Information Engineering, Tiangong University, Tianjin 300387, China; 4School of Mechanical Engineering, Hebei University of Technology, Tianjin 300401, ChinaG.Feng@hebut.edu.cn (G.F.); 5School of Software, Tiangong University, Tianjin 300387, China; sunshengke@tiangong.edu.cn

**Keywords:** fault diagnosis, bearing compound fault, GDDCNN, CReLU, GMP

## Abstract

This study researched the application of a convolutional neural network (CNN) to a bearing compound fault diagnosis. The proposed idea lies in the ability of CNN to automatically extract fault features from complex raw signals. In our approach, to extract more effective features from a raw signal, a novel deep convolutional neural network combining global feature extraction with detailed feature extraction (GDDCNN) is proposed. First, wide and small kernel sizes are separately adopted in shallow and deep convolutional layers to extract global and detailed features. Then, the modified activation layer with a concatenated rectified linear unit (CReLU) is added following the shallow convolution layer to improve the utilization of shallow global features of the network. Finally, to acquire more robust features, another strategy involving the GMP layer is utilized, which replaces the traditional fully connected layer. The performance of the obtained diagnosis was validated on two bearing datasets. The results show that the accuracy of the compound fault diagnosis is over 98%. Compared with three other CNN-based methods, the proposed model demonstrates better stability.

## 1. Introduction

With the development of modern industry, machine health monitoring becomes more and more important to maintain the safe operation of modern mechanical equipment [1,2]. Bearings are an indispensable part of modern machinery, especially in rotating machinery. Rolling bearing faults account for a large proportion of mechanical equipment faults, so it is necessary to ensure the normal operation of bearings [3,4,5]. In practical engineering, because of the complex and changeable operating conditions, once a fault occurs, it often affects other parts, leading to a compound fault [6,7,8]. Compared with single faults, the vibration signals are affected by mutual coupling and interference from various fault features, so the difficulty of feature extraction and fault diagnosis is greatly increased [9]. An effective and reliable compound fault diagnosis of rolling bearings is of great significance in guaranteeing safe operation.

In recent years, the compound fault diagnosis of rolling bearings has received increasing interest and attention from researchers. The research methods for compound fault diagnosis mainly include compound fault mechanism research [10,11], the blind source separation algorithm [12,13,14,15], the signal decomposition algorithm [16,17], and the artificial intelligence algorithm [18,19]. The method based on the compound fault mechanism, taking specific machineries as research objects, limits the application and portability of the model. Methods based on blind source separation have high requirements for the number of channels of a raw signal. The number of sensors must meet the requirements of the algorithm, which increases the cost of a fault diagnosis. The modal decomposition algorithm is a typical signal decomposition method that is suitable for non-stationary signal processing.

However, there are some issues, such as mode aliasing and the end effect, that can directly affect the accuracy of a compound fault diagnosis. With the rapid development of artificial intelligence techniques, machine learning methods, including support vector machines (SVMs), Bayesian classifiers, artificial neural networks (ANNs), and convolutional neural networks (CNNs), have already been applied as potential tools to fault diagnosis, especially deep learning approaches. Ref. [20] proposed a deep inception net with atrous convolution and applied this model to a bearing fault diagnosis. This model overcomes the problem caused by the different feature distributions of characteristics between two data sets and achieves high accuracy. Ref. [21] proposed a model based on CNN, named WDCNN, for a bearing fault diagnosis that achieves high accuracy on raw vibration signals directly in the Case Western Reserve University (CWRU) bearing data set. Ref. [22] utilized an improved CNN for a fault diagnosis. In that network, a multiscale cascaded layer is added to CNN, which can enhance the classification information of the input. Ref. [23] constructed a CNN with feature alignment that addressed the finite-shift-invariance problem, which can extract a robust fault feature. Ref. [24] adopted a multi-task CNN through information fusion for a fault diagnosis on two bearing data sets. The experiment results showed that the proposed model improved the accuracy of the fault diagnosis. Ref. [25] implemented a lightweight CNN that combined transfer learning and self-attention, which can achieve higher diagnosis accuracy than traditional CNN models. Ref. [26] proposed a novel multiscale CNN model that incorporates multiscale learning during the feature extraction process to diagnose the fault of a wind turbine gearbox. A novel multiscale residual attention CNN model was proposed in [27], which utilizes multiscale features, an attention mechanism, and residual learning to enhance feature extraction ability. Experimental validation on two bearing datasets demonstrated that the algorithm achieved higher accuracy. Based on a CNN model, [28] designed a multi-task CNN model that utilizes a speed identification task and a load identification task as auxiliary tasks to improve the performance of a fault diagnosis task. The experimental results showed that multi-task learning can enhance the fault diagnosis performance of the model. In [29], an end-to-end fault diagnosis model combining CNN with LSTM is designed, which can realize a bearing fault diagnosis in a short time. Ref. [30] presented an improved one-dimensional multiscale model that combined different extended convolutional kernels with varying dilation rates. The superiority of this approach was validated on the CWRU and PU datasets. Ref. [31] integrated vibration signals and sound signals through a one-dimensional CNN for fusion and validated that it has a higher diagnostic accuracy. Ref. [32] used short-time Fourier transform to convert vibration signals into spectrograms and adopted a model based on CNN for feature extraction and health status classification. Ref. [33] proposed a lightweight CNN combined with data augmentation technology for a bearing fault diagnosis. A novel hybrid CNN-MLP model for diagnoses was proposed in [34], which combined mixed input to achieve a rolling bearing diagnosis. In [35], a lightweight CNN model with fixed feature graph dimensions is constructed with down-sampling vibration signals to construct spectral graphs, which can achieve high classification accuracy on low-dimensional input data. Ref. [36] put forward a model based on optimized-parameter maximum-correlated kurtosis deconvolution and CNN for a bearing compound fault diagnosis and verified its effectiveness.

The above models demonstrate that deep learning approaches have significantly improved fault diagnosis effectiveness. However, the relationship between compound fault and single fault is one-to-many or many-to-many, which is not a simple superposition of a single fault signal. Compared with signal faults, feature extraction and location of compound faults are more difficult and challenging, which brings great difficulties to compound fault diagnosis. Currently, intelligent diagnosis methods based on deep learning mainly focus on single faults. Some models are suitable for single fault diagnosis; however, they are not effective for compound fault diagnosis. The research and application of deep learning approaches to bearing compound fault diagnosis are still in their infancy. Thus, in this study, a novel deep convolutional neural network combining global feature extraction with detailed feature extraction (GDDCNN) was proposed. 

The proposed GDDCNN model is a deep convolutional neural network that incorporates both global feature extraction and detailed feature extraction, where G means global feature extraction, D means detailed feature extraction, and DCNN represents a deep convolutional neural network. DCNN is a feature-progressive learning algorithm in which the deep network can continue to learn more advanced fault features based on shallow features. By designing two feature extraction modules, G and D, better learning ability can be achieved for DCNN. Therefore, more abundant fault features can be extracted through GDCNN, thus improving the performance of fault diagnosis. The contributions of this study are summarized as follows:A novel deep convolutional neural network combining global feature extraction with detailed feature extraction (GDDCNN) is proposed to extract features adaptively from a raw signal.The modified activation concatenated ReLU (CReLU) is applied in the shallow layer of GDDCNN. It can improve the performance of global feature extraction.The global max pooling (GMP) strategy is designed to replace the traditional fully connected layer, which can extract shift-invariance features and reduce model parameters. It overcomes the overfitting problem in the training process.

The rest of this study is described as follows. Section 2 reviews the basic theory of CNN. Section 3 expresses the proposed model in detail. Section 4 presents the experimental validation through two bearing datasets. Finally, Section 5 draws the conclusion.

## 2. Theoretical Background

CNN can be built with multiple layers, comprising a convolution layer, a pooling layer, an activation layer, and a classification layer. The convolution layer, pooling layer, and activation layer are used to extract features from input signals. One-dimensional CNN has been widely applied to 1-D vibration signal processing due to its powerful feature extraction ability. The classification layer applies the extracted features to classify. 

The convolution layer is the core layer of the CNN structure, which convolves the input data with filter kernels. The network makes the filter learn to activate when it extracts certain features, then realizes feature extraction. The mathematical form can be described as follows:(1)$$y_{j}^{l}=K_{i}^{l}*x_{i}^{l}=\sum_{i\in M_{j}}x_{i}^{l}*w_{ij}^{l}+b_{j}^{i}$$
where the $y_{j}^{l}$ denotes the output of the *l*-th layer; $K_{i}^{l}$ is the *i*-th convolution kernel of the *l*-th layer; $x_{i}^{l}$ denotes the input of the *l*-th layer; the notation * represents the convolution operation; $w_{ij}^{l}$ denotes the weights the convolution kernel; and $b_{j}^{i}$ is the offset.

The activation layer is usually followed by the convolution layer, which is an essential layer. The activation function defines the output and input connections of a neuron, which is usually nonlinear. It makes the network learn nonlinear features from an input vibration signal to improve the capability of feature extraction. Rectified linear unit (Re*LU*) [37] is commonly used as an activation function in CNN, which is defined as follows:(2)ReLU(yjl)=0 yjl<0yjl yjl≥0
which denotes the activation value is 0 on the negative half-axis.

Batch normalization (BN) [38] is applied to deep neural networks, which can reduce the shift of internal covariance and improve the accuracy of the training model. In addition, BN compels the learned options into a standard distribution with a mean value of 0 and a variance of 1, which can accelerate the training speed of the model. The transform of BN is described as follows:(3)$$\left\{\begin{array}{l} \mu_{B}=\frac{1}{m}\sum_{i=1}^{m}x_{i}\\ \delta_{B}^{2}=\frac{1}{m}\sum_{i=1}^{m}(x_{i}-\mu_{B}{)}^{2}\\ {\hat{x}}_{i}=\frac{x_{i}-\mu_{B}}{\sqrt{\delta_{B}^{2}+\varepsilon }}\\ y_{i}=\gamma {\hat{x}}_{i}+\beta \end{array}\right.$$
where *m* represents the mini-batch size, $\mu_{B}$ expresses the mini-batch mean, and $\delta_{B}^{2}$ represents the mini-batch variance. *γ* and *β* are learnable parameters of the network.

The pooling layer performs a down-sampling operation, which removes redundant features and extracts deeper features. Max pooling and average pooling are the most common pooling operations. Max pooling generally outperforms average pooling for time series classification tasks, which is expressed as follows:(4)$$p_{i}^{l}=\max_{(j-1)s+1\leq t\leq js}\{a_{i}^{l}(t)\}$$
where the $p_{i}^{l}$ represents the output features of the *l*-th layer; max is max pooling, $a_{i}^{l}(t)$ means the output value of the *t*-th neuron in the *i*-th channel of the *l*-th layer, *s* denotes the stride of the pooling.

In the classification layer, the Softmax function is applied to normalize the probability of each category in the output. Softmax in the neural network is defined as
(5)$$\mathrm{Softmax}(x_{i})=\frac{e^{x_{i}}}{\sum_{j=1}^{K}e^{x_{j}}}$$
where *K* is the number of categories and $x_{i}$ represents the logits of the *j*-th output neuron.

## 3. Proposed GDDCNN

To overcome the problems of compound fault feature extraction, a novel deep CNN is proposed. The structure of the proposed model, GDDCNN, is shown in Figure 1, composed of an input module, a feature extractor module, a GMP layer, and a Softmax classifier. Global feature extraction and detailed feature extraction constitute the feature extractor module. Besides, this study used two strategies: modified activation operation and GMP strategy during the feature extraction process to enhance the feature extraction ability of the proposed model, GDDCNN. Finally, a compound fault diagnosis is implemented. More details are illustrated in subsequent parts.

### 3.1. GDDCNN Architecture Design

Different kernel sizes for convolution: In CNN, convolutional kernel size plays an important role in the convolutional layer because kernels of different sizes can obtain different features. Generally, wider kernels pay more attention to global information during convolution operations, thereby extracting more global features, while smaller kernels can capture more detailed features. To obtain more robust features from a raw vibration signal, global feature extraction and detailed feature extraction are combined and applied to the feature extractor module. This study designed different kernel sizes for convolution; wide kernel sizes are applied in the shallow (first and second) convolution layers to extract global features, while the deep convolutional kernels are small, which help to obtain detailed features. Multi-convolution layers that adopt small convolutional kernels make the CNN networks deeper, which can improve the performance of compound fault feature extraction. Finally, the size of the convolution kernel for different layers is set to be [64, 16, 3].

Modified activation operation: In addition to the convolution, the activation operation also affects the performance of CNN. Due to the advantages of simple calculation and no vanishing gradient problem, ReLU is widely applied in CNN as an activation unit. However, when the inputs are negative, the neurons are always inactive. These dead neurons in a network may never activate, which stops learning and thus affects the learning ability of the network. In [39], it was found that in the shallow layer of CNNs, the parameter distribution of the network exhibits a stronger negative correlation, while with the deepening of the network, this negative correlation gradually becomes weak. CReLU is a concatenated ReLU, which, through inverting the feature map to activate the negative inputs, helps features transmit better backward. CReLU is defined as follows:(6)CReLU(x)=[ReLU(x),ReLU(−x)]
where Re*LU* represents the Re*LU* activation function. The modified activation CReLU is used in the shallow (first and second) convolution layers, which can improve the performance of global feature extraction.

GMP strategy: The fully connected layer is generally applied after the last convolutional or pooling layer, which can integrate the class-differentiated local features extracted by CNN. Each neuron in the fully connected layer is fully connected to all the neurons in the previous layer. Due to its fully connected characteristics, the parameters of the fully connected layer are numerous, and the calculation is extremely complicated. The right part of Figure 2 shows the GMP progress, that is, the max value of each channel is used as the new feature vector. The GMP strategy clearly reduces the dimension of the feature vector, which can avoid overfitting. Another advantage is that it is more native to the convolution structure by enforcing correspondences between feature maps and categories. Furthermore, this strategy can retain the spatial information that keeps shift-invariance, resulting in robust extracted features.

### 3.2. Training of GDDCNN

The architecture of GDDCNN is designed to take advantage of one-dimensional CNN. The cross-entropy loss function is used to estimate the consistency between the Softmax output probability distribution and the target class probability distribution. Suppose that *p*(*x*) and *q*(*x*) represent the target distribution and estimated distribution, respectively. The loss function can be expressed as follows:(7)$$Loss(L)=H(p(x),q(x))=-\sum_{x}p(x)logq(x)$$

The gradient of loss L and the gradient of parameters related to BN should be back-propagated in GDDCNN during training as per the following rules:(8)$$\frac{\partial L}{\partial {\bar{x}}_{i}}=\frac{\partial L}{\partial y_{i}}\cdot \gamma$$
(9)$$\frac{\partial L}{\partial \delta_{B}^{2}}=\sum_{i=1}^{m}\frac{\partial L}{\partial {\bar{x}}_{i}}\cdot (x_{i}-\mu_{B})\cdot \frac{-1}{2}{\left(\delta_{B}^{2}+\varepsilon \right)}^{-3/2}$$
(10)$$\frac{\partial L}{\partial \mu_{B}}=(\sum_{i=1}^{m}\frac{\partial L}{\partial {\bar{x}}_{i}}\cdot \frac{-1}{\sqrt{\delta_{B}^{2}+\varepsilon }})+\frac{\partial L}{\partial \delta_{B}^{2}}\cdot \frac{\sum_{i=1}^{m}-2(x_{i}-\mu_{B})}{m}$$
(11)$$\frac{\partial L}{\partial x_{i}}=\frac{\partial L}{\partial {\bar{x}}_{i}}\cdot \frac{-1}{\sqrt{\delta_{B}^{2}+\varepsilon }}+\frac{\partial L}{\partial \delta_{B}^{2}}\cdot \frac{2(x_{i}-\mu_{B})}{m}+\frac{\partial L}{\partial \mu_{B}}\cdot \frac{1}{m}$$
(12)$$\frac{\partial L}{\partial \gamma }=\sum_{i=1}^{m}\frac{\partial L}{\partial y_{i}}\cdot {\bar{x}}_{i}$$
(13)$$\frac{\partial L}{\partial \beta }=\sum_{i=1}^{m}\frac{\partial L}{\partial y_{i}}$$

Adam is a learning rate adaptive optimization algorithm that assembles the Adagrad algorithm and the RMSProp algorithm, which allows a model to allocate more updates to rarely occurring features, thereby aiding in the convergence of the optimization process. By bringing together the adaptive learning rates of Adagrad and the stability of RMSProp, the Adam algorithm provides an effective solution to optimize the proposed model, enabling it to converge quickly and efficiently while avoiding common optimization pitfalls. The choice of optimizer plays a crucial role in the success of the proposed model’s training process.

In order to minimize the value of the loss function, the Adam optimization algorithm is applied to update the weights and obtain the optimal weights. The Adam optimizer is chosen for the following reasons: Bearing vibration signals typically contain a large number of data points due to their time-series nature. The efficient convergence property of the Adam algorithm allows the model to learn from a large volume of data more quickly. This, in turn, accelerates the assessment of the bearing’s health status, saving training time. Features in bearing vibration signals may vary in importance over different time intervals. Adam’s adaptive learning rate automatically adjusts the learning rate based on the gradient of each parameter. This capability helps the model better adapt to dynamic variations within a signal. When the Adam optimizer is initialed, it is necessary to set the learning rate ε, exponential decay rate of moment estimation $\rho_{1},\rho_{2}$, and constant δ. In this experiment, we set ε to 0.001, $\rho_{1},\rho_{2}$ to default 0.9, 0.999, and the constant was set to prevent the numerical mutation from being set to $10^{-8}$ during the dividing operation. The detailed process of the Adam algorithm is shown in Table 1. More details on the Adam algorithm can be found in [40].

### 3.3. Diagnosis Procedure

The raw vibration signals of rolling bearings are collected by a data acquisition device.The obtained vibration signals are sliced into samples of length 2048 for standardization processing and then used as network input.Extracting fault features by combining global feature extraction with detailed feature extraction. Utilizing the GMP layer to integrate features. Then, Softmax is employed as a classifier to classify fault features.Testing samples are entered into the network to realize fault diagnosis and validate the performance of the proposed model, GDDCNN.

## 4. Experiments and Results

### 4.1. Data Acquisition

In this study, the two-stage gear drive test bench shown in Figure 3, is composed of a drive motor, a two-stage helical gear box, a powder brake, a torque speed sensor, a coupling, tested bearings, and a data acquisition device. The experimental data was obtained by the intermediate shaft end bearing of the two-stage helical gearbox. In this task group, three rotating speeds (1200 rmp, 1500 rmp, and 1800 rmp) and three electrical machinery loads (0 A, 0.5 A, and 1.0 A) were set up in 9 working conditions in the experiment, respectively. In addition, the sampling frequency is 96 kHz, and the sampling time is 10 s. Collecting 9 raw data samples from each category, with a sampling length of 96,000 for each raw data sample. There are six categories of bearing faults: one normal, three single faults, and two compound faults. These six health conditions of bearings are designated as normal (NO), inner fault (IF), outer fault (OF), roll fault (RF), inner and roll compound fault (IRF), and outer and roll fault (ORF). All bearing faults simulated by electrical discharge machining (EDM) technology are shown in Figure 4. 

In addition to the data obtained from the above two-stage gear drive test bench, this study also utilized the public Case Western Reserve University (CWRU) dataset [41] to simulate compound fault signals. The raw vibration signals under a load of 1 hp with a rotating speed of 1772 r/min and a sampling frequency of 12kHz are selected. Four compound fault categories are simulated by selected signals, which are IF + RF, OF + RF, IF + OF, and IF + OF + RF.

### 4.2. Data Preprocessing

*(1) Sample Segmentation:* After data acquisition, an easy data augmentation technique of slicing the raw vibration signals with overlap is applied to process training samples. In order to reflect the periodicity of raw signals, this study set the sampling size to 2048 and the overlapping window size to 512. The data segmentation process is shown in Figure 5. After data segmentation, 16,848 samples of each category were obtained, totaling 101,088 samples.

*(2) Standardization:* After sample segmentation, in order to improve the convergence speed and accuracy of the model, the samples need to be standardized. The samples are standardized by the z-score method and follow the distribution of the mean value 0 and variance 1. The standardization process for the raw data ($x_{1,}$$x_{2,}x_{3}\ldots x_{n}$) is as follows:(14)$$\bar{x}=\frac{1}{n}\sum_{i=1}^{n}x_{i}$$
(15)$$\delta =\sqrt{\frac{1}{n-1}\sum_{i=1}^{n}(x_{i}-\bar{x}{)}^{2}}$$
(16)$$\mathrm{Standard}(x_{i})=\frac{x_{i}-\bar{x}}{\delta }$$

### 4.3. Parameters of the Proposed Network

The experiments were performed using the above data to verify the performance of the proposed method. The architecture of the proposed method involves stacking 2 convolutional layers with a wide kernel size and pooling layers, 4 convolutional layers with a small kernel size and pooling layers, followed by GMP and a Softmax layer to implement fault diagnosis. The first convolutional kernel size and channel depth are, respectively, set to 64 and 16; the second convolutional kernel size and channel size are set to be 16 and 32, respectively; and the rest of the convolutional kernel size and channel are set to be 3 and 64, respectively. All five categories of pooling are max pooling, with a size of 2. The first two activation functions are the improved activation connection ReLU (CReLU), and the remaining activation functions are ReLU. To improve the performance of the proposed model, GDDCNN, BN is applied after each activation layer. Each convolution layer and pooling layer utilizes a zero-padding of causal and same type to prevent the loss of edge information. The detailed parameters of the proposed network are listed in Table 2.

### 4.4. Comparative Experiment

In order to validate the feasibility of the proposed method, which is compared with several currently popular models, including classical CNN, CNN-SVM, and WDCNN [21]. This study selected 1500 samples from each category under all working conditions randomly for experiments: 1000 samples as training data, 300 samples as verification data, and 200 samples as testing data. The details of the experimental dataset are shown in Table 3. All experiments were run on a server equipped with an Intel Core i7-6800k CPU, an Nvidia GeForce GTX 1080 Ti GPU, and 32G of RAM.

Accuracy is the most common metric used to evaluate the performance of a classifier. It calculates the proportion of correctly classified samples out of the total number of samples. Specifically, accuracy can be calculated using the following formula:(17)$$Accuray=\frac{TP+TN}{TP+TN+FP+FN}$$
where *TP* represents the number of true positive samples, i.e., the number of samples correctly predicted as positive by the classifier; *TN* represents the number of true negative samples, i.e., the number of samples correctly predicted as negative by the classifier; *FP* represents the number of false positive samples, that is, the number of samples incorrectly predicted as positive by the classifier; and *FN* represents the number of false negative samples, that is, the number of samples incorrectly predicted as negative by the classifier.

The results of the diagnosis accuracy for each category using four different methods are summarized in Table 4. According to the statistical results, it can be observed that the single fault diagnosis accuracy of the proposed method can reach over 98% or even approach 100%, and the compound fault diagnosis accuracy is also over 95%. The accuracy of other methods in category 4 (IRF) of the compound fault diagnosis is lower than 90%.

In order to demonstrate the advantages of the proposed method more visually and evidently, a line chart is used to compare the accuracy of the four methods, as shown in Figure 6. Where fault category 0–5 indicates a fault label, the details about fault labels correspond to the fault category, as shown in Table 3. It can be clearly observed that the proposed method has better performance than the other three methods. The diagnosis accuracy of each category is higher than that of other methods. Moreover, the diagnosis accuracy of each category is relatively balanced, with none of the six categories having particularly low accuracy. Although the WDCNN model has higher accuracy than CNN and CNN-SVM, its performance and stability are still not comparable to the proposed method.

Precision is similar to but different from accuracy, which is the proportion of correctly predicted positive samples among all samples identified as positive. Precision represents the prediction accuracy of the positive sample results. Recall refers to the proportion of correctly predicted positive samples among all true positive samples, which reflects the proportion of positive samples correctly predicted. F1-Score is the harmonic mean of precision and recall, which combines information from these two metrics. The calculation of precision, recall, and F1-score is as follows:(18)$$Precision=\frac{TP}{TP+FP}$$
(19)$$Recall=\frac{TP}{TP+FN}$$
(20)$$F1-Score=\frac{2\times Precision\times Recall}{Precision+Recall}$$
in which TP, FP, and FN have the same meanings as those in Formula 17. The precision, recall, and F1-score of the proposed method are shown in Figure 7. Where Class 0 represents normal, Class 1, Class 2, and Class 3 represent inner fault, outer fault, and roll fault. Both Class 4 and Class 5 are compound faults; Class 4 expresses inner and roll compound faults, while Class 5 represents outer and roll faults.

Precision, recall, and F1-score are also crucial performance metrics for a classifier; a higher precision, recall, and F1-score indicates a better performance of the classifier. It can be observed from Figure 7 that all the values of the proposed model for all conditions are high.

In order to further validate the effectiveness of the fault diagnosis model, three error metrics—maximum average error (MAE), mean squared error (MSE), and root mean squared error (RMSE)—are selected to evaluate the fitting effect of the fault diagnosis model. The formulas for each metric are defined as follows:(21)$$MAE=\frac{1}{N}\sum_{i=1}^{N}\left|Y_{i,true}-Y_{i,pred}\right|$$
(22)$$MSE=\frac{1}{N}\sum_{i=1}^{N}{\left(Y_{i,true}-Y_{i,pred}\right)}^{2}$$
(23)$$\begin{array}{l} RMSE=\sqrt{\frac{1}{N}\sum_{i=1}^{N}{\left(Y_{i,true}-Y_{i,pred}\right)}^{2}} \end{array}$$
where *N* is the number of testing samples, the value of $Y_{i,true}$ is 1, and $Y_{i,pred}$ is the probability value of the category to which it belongs. The value of $Y_{i,pred}$ calculated through the Softmax classifier is closer to 1, indicating a better fitting effect of the model. Therefore, smaller values of the three metrics represent better stability for the model. The metrics results of the four methods are listed in Table 5. From the statistical results, it can be seen that the three metrics of the GDDCNN are all smaller than those of other methods, indicating that the GDCNN proposed in this study has good stability.

Finally, the feature distribution in different layers of the proposed model, GDDCNN, can be visualized through the t-SNE technique. t-SNE is a nonlinear dimensionality reduction algorithm that calculates the positions of sample points based on the similarity between sample points in high-dimensional space and the distances between sample points in low-dimensional space. The experiment results, i.e., the position information after dimension reduction of the raw signal, six convolutional layers, and the GMP layer, are shown in Figure 8. It can be obviously seen that the raw signal and the features in the shallow convolution layers are inseparable. With the deepening of the convolution layer, the feature distribution can be improved significantly. From the visualization of the GMP layer, we can observe the feature distribution, which suggests that the proposed model cannot discriminate between category 1 and category 4 very well. It may be due to the fact that category 4 is a compound fault that contains both category 1 and category 3. The samples of other categories can cluster in their own area, and the inter-category differences are good. The experiment results indicate that the proposed model, GDDCNN, can extract robust and effective features from a raw signal and achieve efficient fault diagnosis.

In order to further verify the effectiveness and performance of the proposed model, GDDCNN, fault diagnosis experiments are carried out with the simulated data from the CWRU dataset. Data is processed by overlapping sampling during simulation. The distribution of samples is listed in Table 6.

The average accuracy of the four models is shown in Table 7. It is easy to see from the results that all models achieve high diagnostic accuracy. The accuracy of the proposed model, GDDCNN, reaches 100%, which indicates it has good performance.

## 5. Conclusions

This study proposed a novel deep CNN named GDDCNN for the diagnosis of practical engineering compound faults. GDDCNN extracted robust features adaptively from raw vibration signals and realized compound fault diagnosis without any manual processing. The datasets acquired by a two-stage gear drive test bench and simulated with the published dataset of CWRU were applied to experiments to verify the performance of the proposed model. The experimental results demonstrate that GDDCNN can achieve more than 98% accuracy on both datasets. Compared with three other existing CNN-based models, this model not only achieves satisfactory diagnostic accuracy but also exhibits strong stability. By combining different convolutional kernel sizes, improved activation CReLU, and GMP strategy, the performance of GDDCNN has been improved, and compound fault diagnosis has been achieved more effectively. In further work, we will continue to explore how to adaptively select hyperparameters to enhance the accuracy and stability of the diagnosis model. We also aim to research the application of transfer learning to achieve compound fault diagnosis across different operating conditions.

## Figures and Tables

**Figure 1 sensors-23-08060-f001:**
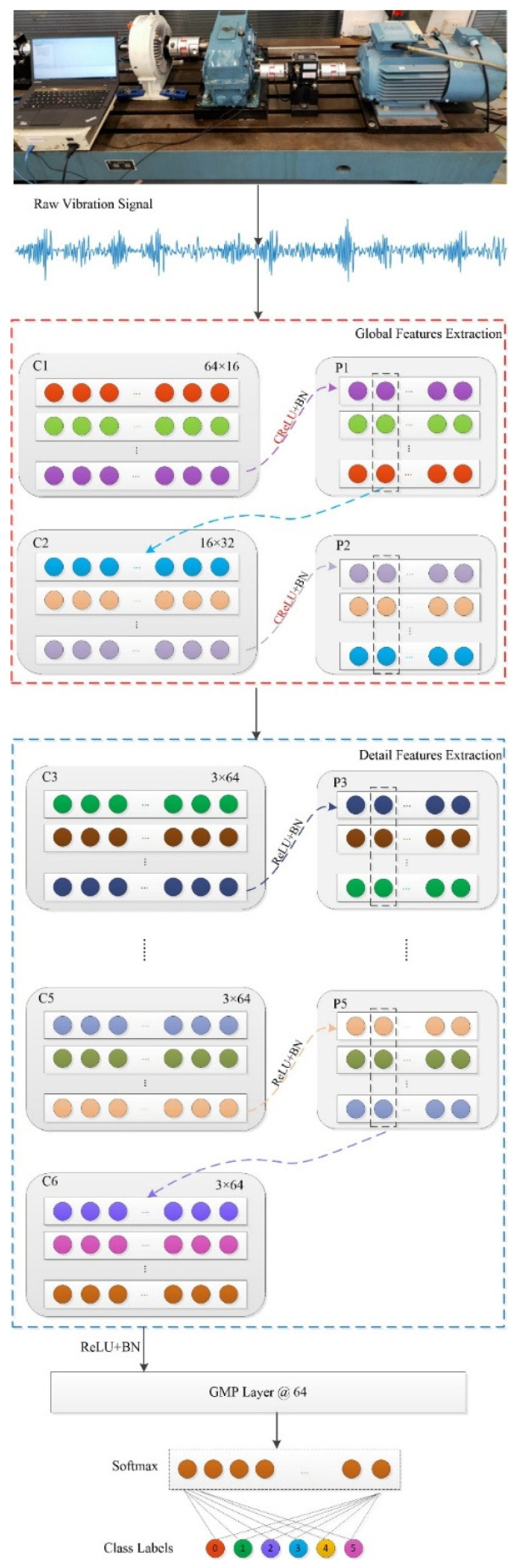
Structure of GDDCNN.

**Figure 2 sensors-23-08060-f002:**
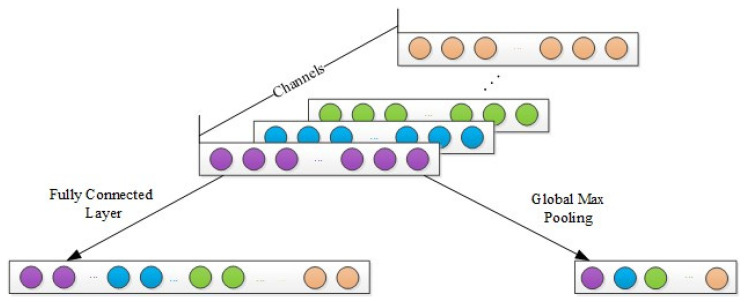
Global max pooling.

**Figure 3 sensors-23-08060-f003:**
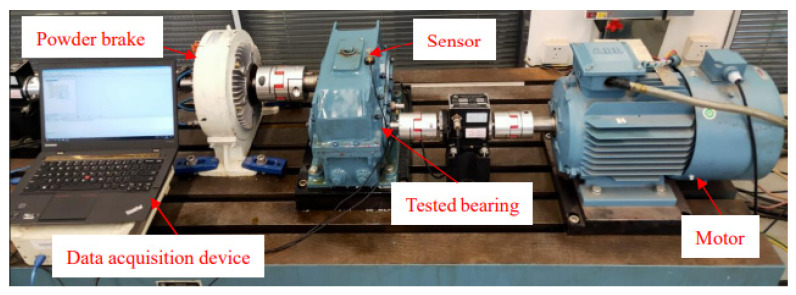
Two-stage gear drive test bench.

**Figure 4 sensors-23-08060-f004:**
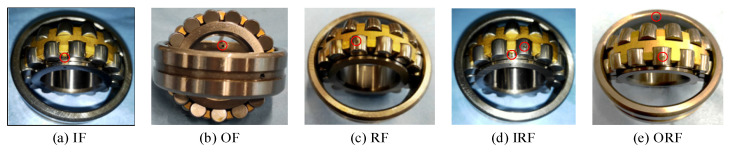
Bearing fault simulation: (**a**) inner fault; (**b**) outer fault; (**c**) roll fault; (**d**) inner and roll compound fault; and (**e**) outer and roll fault.

**Figure 5 sensors-23-08060-f005:**
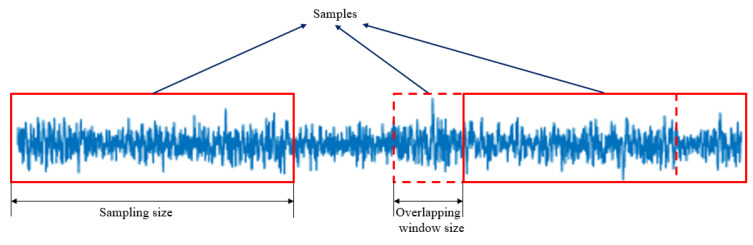
Data segmentation process.

**Figure 6 sensors-23-08060-f006:**
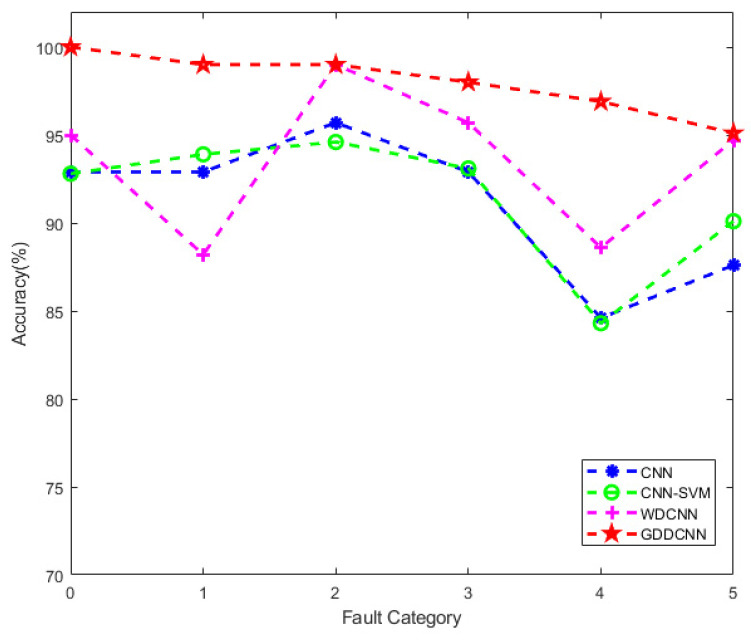
Accuracy comparison of four methods.

**Figure 7 sensors-23-08060-f007:**
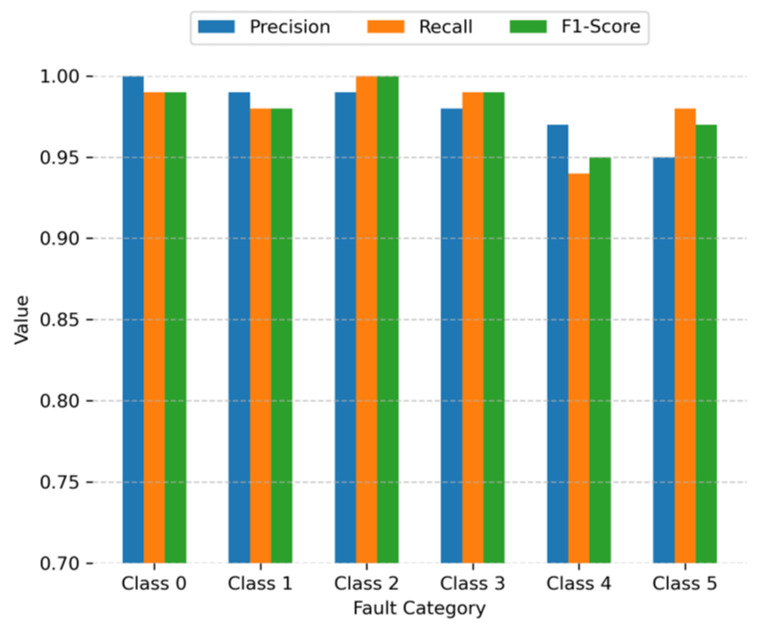
Precision, recall, and F1-score of the GDDCNN.

**Figure 8 sensors-23-08060-f008:**
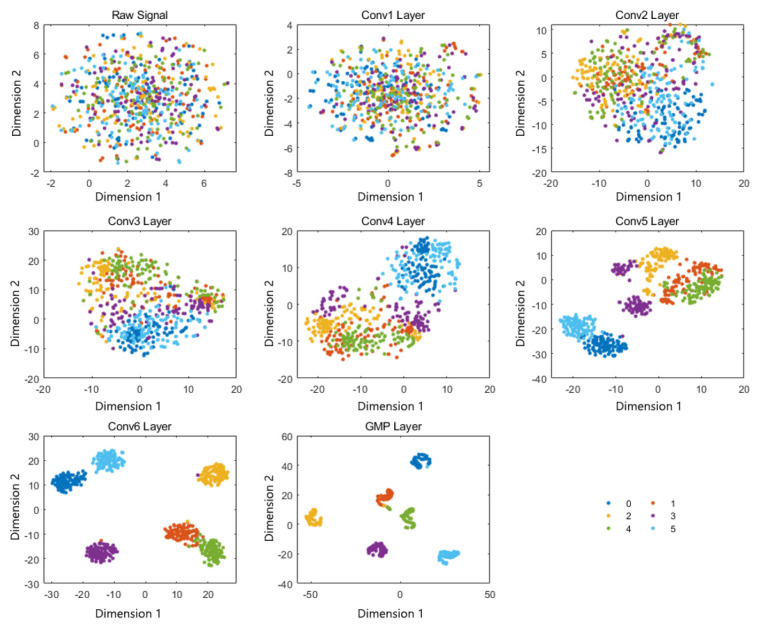
Feature visualization via t-SNE.

**Table 1 sensors-23-08060-t001:** Detailed flow of the Adam algorithm.

Adam Algorithm
Initialize variables $\varepsilon ,\rho_{1},\rho_{2},\delta ,\theta ,s=0,r=0,t=0$
**While** stopping criterion not met **do**
Sample a minibatch of m examples from the training set
$\left\{x^{\left(1\right)},\ldots ,x^{\left(m\right)}\right\}$ with corresponding targets y(i)
Computing gradient:
$g\leftarrow \frac{1}{m}\nabla_{\theta }\sum_{i}L(f(x^{\left(i\right)};\theta ),y^{\left(i\right)})$
t←t+1
Update biased first-order moment estimation: $s\leftarrow \rho_{1}s+(1-\rho_{1})g$
Update biased second-order moment estimation: $r\leftarrow \rho_{2}r+(1-\rho_{2})g⊙g$
Correcting the deviation of the first moment: $\hat{s}\leftarrow \frac{s}{1-\rho_{1}^{t}}$
Correcting the deviation of the second moment: $\hat{r}\leftarrow \frac{r}{1-\rho_{2}^{t}}$
Calculation update: $\Delta \theta =-\varepsilon \frac{\hat{s}}{\sqrt{\hat{r}}+\delta }$
Apply update: θ←θ+Δθ
**end while**

**Table 2 sensors-23-08060-t002:** Detailed parameters of the proposed network.

Layer Type	Kernel Size/Stride	Kernel Channel Size	Padding	Output Size
Input	/	/	/	2048
C1	64/8	16	causal	256*16
P1	2/2	/	same	128*32
C2	16/2	32	causal	64*32
P2	2/2	/	same	32*64
C3	3/1	64	causal	32*64
P3	2/2	/	same	16*64
C4	3/1	64	causal	16*64
P4	2/2	/	same	8*64
C5	3/1	64	causal	8*64
P5	2/2	/	same	4*64
C6	3/1	64	same	4*64
GMP	/	/	/	64*1
Softmax	/	/	/	6*1

**Table 3 sensors-23-08060-t003:** Description of samples distribution.

Fault Location	Sample Number	Training/Validation/Test	Label
Normal	1500	1000/300/200	0
IF	1500	1000/300/200	1
OF	1500	1000/300/200	2
RF	1500	1000/300/200	3
IRF	1500	1000/300/200	4
ORF	1500	1000/300/200	5

**Table 4 sensors-23-08060-t004:** Classification accuracy results of the four methods.

Method	Class Label					
	0	1	2	3	4	5
CNN	92.9%	92.9%	95.7%	92.9%	84.6%	87.6%
CNN-SVM	92.8%	93.9%	94.6%	93.1%	84.3%	90.1%
WDCNN	95%	88.2%	99%	95.7%	88.6%	94.7%
GDDCNN	100%	99%	99%	98%	96.9%	95.1%

**Table 5 sensors-23-08060-t005:** Metrics results of the four methods.

Method	MAE	MSE	RMSE
CNN	0.265	0.861	0.928
CNN-SVM	0.270	0.860	0.927
WDCNN	0.108	0.275	0.524
GDDCNN	0.032	0.065	0.255

**Table 6 sensors-23-08060-t006:** Distribution of samples.

Fault Location	Sample Number	Training/Validation/Test	Label
Normal	1000	700/200/100	0
IF	1000	700/200/100	1
OF	1000	700/200/100	2
RF	1000	700/200/100	3
IF + RF	1000	700/200/100	4
OF + RF	1000	700/200/100	5
IF + OF	1000	700/200/100	6
IF + OF + RF	1000	700/200/100	7

**Table 7 sensors-23-08060-t007:** Average accuracy of the four models.

Model	CNN	CNN-SVM	WDCNN	GDDCNN
Accuracy (%)	98	98	98.75	100

## Data Availability

Not applicable.

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
