# Peer review of "A Novel Deep Convolutional Neural Network Combining Global Feature Extraction and Detailed Feature Extraction for Bearing Compound Fault Diagnosis"

_sensors, 2023, doi:10.3390/s23198060_

Round 1
Reviewer 1 Report
The application is interesting, but the adopted algorithm is computationally low when compared to competitive algorithms.
1. The finding of the algorithms should be mentioned in the abstract section.
2. Abbreviations are pathetic, provide separate section for it.
3. Instead of concentrating Bearing Compound Fault Diagnosis, the author try to concentrate on the CNN and its family algorithms. Why?
4. What is meant by subsection in section 3.1. Provide proper title.
5. Related work section should be improved. I recommend the author to include 10-15 recent year papers based on the CNN and bearing fault analysis.
6. IN GDDCNN - What is meant by GDD? I couldn’t find the proper description. Include it.
7. Equation 3 is wrong. Right side operator is missing.
8. What is meant by global features extraction ? How it supported to classify the fault.
9. Equation 7: Mention Loss should be Loss(L).
10. How many samples are generated? Mention it.
11. The author mentioned “In order to minimize the value of the loss function, Adam optimization algorithm is applied to update the weights and obtain the optimal weights.” - The author used adam optimizer to update the weight? If so, what is the novelty here?
12. What kind of materials are used in the research. For the test case, change the materials in the bearing.
13. Apart from table 3, remaining tables and contents are pointless. I recommend the author to include few more performance metrics to show the effectiveness of the proposed CNN model.
Recheck and proofread the paper thoroughly.
Reviewer 2 Report
The paper is quite interesting. Refers to the identification of bearing damage. The authors analyze A Novel Deep Convolutional Neural Network Combining, Global Features Extraction with Detail Features Extraction for, Bearing Compound Fault Diagnosis. The paper is well researched and shows interesting experimental and modeling results.
The abstract should be an objective representation of the article and it must not contain results that are not presented and substantiated in the main text and should not exaggerate the main conclusions.
Please correct the citations in the text. References should be numbered in order of appearance and indicated by a numeral or numerals in square brackets—e.g., [1] or [2,3], or [4–6]. Do not use: Chen et al. 22 constructed…... Guo et al. 23 adopted a multi-task …... Zhong et al. 24 implemented……. Gao et al. 25 put forward a model ……etc.
I recommend expanding the paper a bit to about 16 pages
On figure 4. Bearing fault simulation add markings (a), (b), (c), (d), ….etc…and name of bearing fault
On figure 5……… Explain and describe the fault category in the text, below
Figure 6…… Add a designation and unit for the value axis of the function. Explain the meaning of class0.....class 5
Figure 7. Feature visualization via t-SNE…… Describe the individual axes. What variables do they represent.
Clearly define the terms Accuracy, Precision, etc... in your text
Conclusions should be extended and improved. The conclusions should be character is not only qualitative but also quantitative. They should be supported by facts resulting from the conducted research. They should generalize the acquired knowledge and the results of scientific research. Conclusions should be supported by the obtained numerical results. Conclusions should be greatly expanded. Further research plans in this area can be indicated.
Author Contributions should be changed: For research articles with several authors, a short paragraph specifying their individual contributions must be provided. The following statements should be used “Conceptualization, X.X. and Y.Y.; methodology, X.X.; software, X.X.; validation, X.X., Y.Y. and Z.Z.; formal analysis, X.X.; investigation, X.X.; resources, X.X.; data curation, X.X.; writing—original draft preparation, X.X.; writing—review and editing, X.X.; visualization, X.X.; supervision, X.X.; project administration, X.X.; funding acquisition, Y.Y.
Do not use: „Data curation, Yitong Li; Investigation, Shuzhen Han, Guojin Feng and 355 Shengke Sun; Methodology, Shuzhen Han; Resources, Dong Zhen; Software, Shijie Luo; Validation, 356 Pingjuan Niu; Writing – original draft, Shuzhen Han; Writing – review & editing, Pingjuan Niu and 357 Yitong Li”
I encourage the authors to study the new literature in this field. https://doi.org/10.3390/s20072019;
Correspondence: e-mail@e-mail.com; Tel.: (optional; include country code; if there are multiple corresponding authors, add author initials)
Abstract should be of about 200 words maximum. Currently there are 225 words.
Round 2
Reviewer 1 Report
As per the comment the author revised the paper. So, i recommend this paper for further process.
We can accept this paper in current form